# Zero-shot learning for speech recognition with universal phonetic model

## Abstract

There are more than 7,000 languages in the world, but due to the lack of training sets, only a small number of them have speech recognition systems. Multilingual speech recognition provides a solution if at least some audio training data is available. Often, however, phoneme inventories differ between the training languages and the target language, making this approach infeasible. In this work, we address the problem of building an acoustic model for languages with zero audio resources. Our model is able to recognize unseen phonemes in the target language, if only a small text corpus is available. We adopt the idea of zero-shot learning, and decompose phonemes into corresponding phonetic attributes such as *vowel* and *consonant*. Instead of predicting phonemes directly, we first predict distributions over phonetic attributes, and then compute phoneme distributions with a customized acoustic model. We extensively evaluate our English-trained model on 20 unseen languages, and find that on average, it achieves 9.9% better phone error rate over a traditional CTC based acoustic model trained on English.

## 1 Introduction

Over the last decade, Automatic Speech Recognition (ASR) has achieved great successes in many well-resourced languages such as English, French and Mandarin (Amodei et al., 2016; Xiong et al., 2016; Collobert et al., 2016). On the other hand, speech resources are still sparse for the majority of languages (Lewis, 2009). They cannot thus benefit directly from recent technologies. As a result, there is an increasing interest in building speech recognition systems for low-resource languages. As collecting annotations for low resource languages is expensive and time-consuming, researchers have exploited data augmentation (Ragni et al., 2014; Kanda et al., 2013), articulatory features (Stüker et al., 2003a) and multilingual techniques (Schultz & Waibel, 2001; Vu & Schultz, 2013; Tüske et al., 2013; Dalmia et al., 2018b).

An even more challenging problem is to recognize utterances in a language with zero training data. This task has significant implications in documenting endangered languages and preserving the associated cultures (Gippert et al., 2006). This data setup has mainly been studied for unsupervised speech processing field(Glass, 2012; Jansen et al., 2013; Versteegh et al., 2015; Dunbar et al., 2017; Heck et al., 2017; Hermann & Goldwater, 2018), which typically uses an unsupervised technique to learn representations which can be used towards speech processing tasks, detailed in Section 4. Some of them also deal with the Out-Of-Vocabulary (OOV) problem in the language modeling side of ASR (Maas et al., 2015), or visual-only speech recognition (Stafylakis & Tzimiropoulos, 2018).

In terms of zero-shot learning in speech recognition, which consist of learning an acoustic model without any audio data for a given target language, there has not been much work. Previous works typically consider some amount of audio corpus to be present, because the phonetics of a target language usually differ from existing training languages, especially when there are unseen phonemes which are unavailable in existing language resources. In this work, we aim to solve this problem without even considering any target audio data.

The prediction of unseen objects has also been a long-time critical problem in the computer vision field. For specific object classes such as *faces*, *vehicles* and *cats*, a significant number manually labeled data is usually available, but collecting sufficient data for every object human could recognize is impossible. Zero-shot learning attempts to solve this problem to classify even unseen objects using mid-level side information. For example, *zebra* can be recognized by detecting attributes such

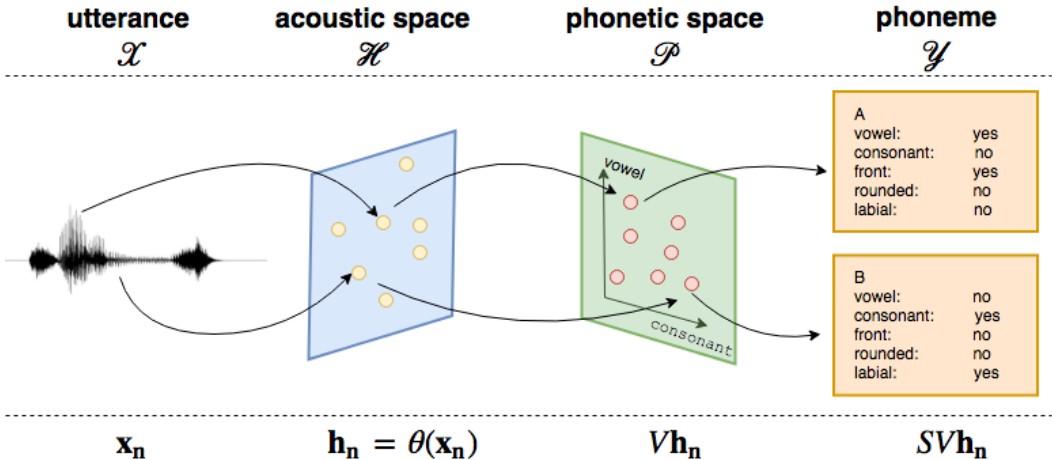

Figure 1: Illustration of the proposed zero-shot learning framework. Each utterance is first mapped into acoustic space $\mathcal{H}$. Then we transform each point in the acoustic space into phonetic space $\mathcal{P}$ with a linear transformation $V$. Finally phoneme distributions can be obtained by applying a signature matrix $S$

as *stripped*, *black* and *white*. Inspired by approaches in computer vision research, we propose the Universal Phonetic Model (UPM) to apply zero-shot learning to acoustic modeling. In this model we decompose the phoneme into its attributes and learn to predict a distribution over various phonetic attributes. This can then be used to infer the unseen phonemes for the test language. For example, the phoneme /a/ can be decomposed into its attributes: *vowel*, *open*, *front* and *unrounded*.

Our approach is summarized in Figure 1. First, frames are extracted and a standard acoustic model is applied to map each frame into the acoustic space $\mathcal{H}$. Next we transform it into the phonetic space $\mathcal{P}$ which reflects the phonetic properties of each frame (such as whether it indicates a vowel or a consonant). Then, we compute the distribution of phonemes for that frame using a predefined signature matrix $S$ which describes relationships between phonetic attributes and phonemes in each language. Finally we adjust the phoneme distribution with a prior distribution estimated from a small set of text corpus (usually 1000 sentences). To the best of our knowledge, this is the first paper applying zero-shot learning for acoustic model in speech recognition without any audio data.

To evaluate our UPM model, we trained our model on English, and evaluated the model on 20 languages. We also trained a standard English acoustic model as a baseline for comparison. The result indicates that we consistently outperform the baseline CTC model, and we achieved 9.9% improvements in phone error rate on average. As we mainly focus on acoustic model in this paper, we adopt phone error rate as our main metric in this work.

The main contributions of this paper are as follows:

1. We propose the Universal Phonetic Model (UPM) that can recognize phonemes that are unseen during training by incorporating knowledge from the phonetics domain.[1]

2. We introduce a sequence prediction model to integrate a zero-shot learning framework for sequence prediction problem.

3. We show that our model is effective for 20 languages, and our model gets 9.9% better phone error rate over the baseline on average.

## 2 APPROACH

This section explains details of our Universal Phonetic Model (UPM). In the first section, we describe how we constructed a proper set of phonetic attributes for acoustic modeling. Next, we

---

[1]Our code would be released upon acceptance

demonstrate how to assign attributes to each phoneme by giving an algorithm to parse X-SAMPA format. Finally we show how we integrate the phonetic information into the sequence model with a CTC loss (Graves et al., 2006).

## 2.1 PHONETIC ATTRIBUTES

Unlike attributes in the computer vision field, attributes of phones are independent of the corpus and dataset, they are well investigated and defined in the domain of articulatory phonetics (Ladefoged & Johnson, 2014). Articulatory phonetics describes the mechanism of speech production such as the manner of articulation and place of articulation, and it tends to describe phones using discrete features such as voiced, bilabial (made with the two lips) and fricative. These articulatory features have been shown to be useful in speech recognition (Kirchhoff, 1998; Stüker et al., 2003b; Müller et al., 2017), and are a good choice for phonetic attributes for our purpose. We provide some categories of phonetic attributes below. The full list of phonetic attributes we used is attached in the Appendix.

**Consonants**. Consonants are formed by obstructing the airstream through the vocal tract. They can be categorized in terms of the place and the manner of this obstruction. The places can be largely divided into three classes: *labial*, *coronal*, *dorsal*. Each of the class have more fine-grained classes. The manners of articulation can be grouped into: *stop*, *fricative*, *approximant* etc.

**Vowel**. In the production of vowels, the airstream is relatively unobstructed. Each vowel sound can be specified by the positions of lips and tongue (Ladefoged & Johnson, 2014). For instance, the tongue is at its highest point in the front of the mouth for *front* vowels. Additionally, vowels can be characterized by properties such as whether the lips are rounding or not (*rounded*, *unrounded*).

**Diacritics**. Diacritics are small marks to modify vowels and consonants by attaching to them. For instance, *nasalization* marks a sound for which the velopharyngeal port is open and air can pass through the nose. To make the phonetic attribute set manageable, we assign attributes of diacritics to some existing consonants attributes if they share similar phonetic property. For example, *nasalization* is treated as the *nasal* attribute in consonants.

In addition to phonetic attributes mentioned above, we note that we also need to allocate an special attribute for blank in order to predict blank labels in CTC model, and backpropagate their gradients into the acoustic model. Thus, our phonetic attribute set $A_{phone}$ is defined as the union of these three domain attributes as well as the blank label.

$$A_{phone} = A_{consonants} \cup A_{vowels} \cup A_{diacritics} \cup \{blank\} \tag{1}$$

### 2.1.1 ATTRIBUTE ASSIGNMENT

---

**Algorithm 1** A simple algorithm to assign attributes to phonemes

---

**input** : X-SAMPA representation of phoneme $p$
**output:** Phonetic attribute set $A \subseteq A_{phone}$ for $p$

$A \leftarrow$ empty set

**while** $p \notin P_{base}$ **do**
    find the longest suffix $p_s \in P_{base}$
    Add $f|_{P_{base}}(p_s)$ to $A$
    Remove suffix $p_s$ from $p$
**end**
Add $f|_{P_{base}}(p)$ to $A$

---

Next, we need to assign each phoneme with appropriate attributes. There are multiple approaches to retrieve phonetic attributes. The most simple one is to use tools to collect articulatory features for each phoneme (Mortensen et al., 2016; Moran et al., 2014). However, those tools only provide coarse-grained phonological features and we expect more fine-grained and customized phonetic features. In this section, we propose a naive but useful approach for attribute assignment. We note that we use X-SAMPA format to denote each IPA (Decker et al., 1999) in this work. X-SAMPA was devised to produce a computer-readable representation for IPA (Wells, 1995). Each IPA segment can

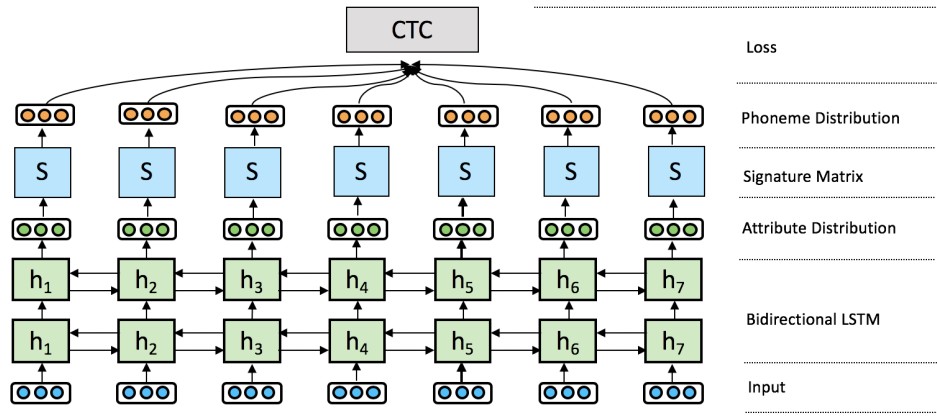

Figure 2: Illustration of the sequence model for zero-shot learning. The input layer is first processed with a Bidirectional LSTM acoustic model, and produces a distribution over phonetic attributes. Then it is transformed into a phoneme distribution by a language dependent signature matrix $S$

be mapped to X-SAMPA with appropriate rule-based tools (Mortensen et al., 2018). For example, IPA /ə/ can be represented as @ in X-SAMPA.

The assignment can be formulated as a problem to construct an assignment function $f : P_{xsampa} \to 2^{A_{phone}}$ where $P_{xsampa}$ is the set of all valid X-SAMPA phonemes. The assignment function should map each phone into its corresponding subset of $A_{phone}$. To construct the function in the entire domain, we first manually mapped a small subset $P_{base} \subset P_{xsampa}$ and constructed a restricted assignment function $f|_{P_{base}}$. The mapping is customizable and has been verified with the IPA handbook(Decker et al., 1999). Then for every $p \in P_{xsampa}$, we continue to remove diacritics suffix from it until it could be find in $P_{base}$. For example, to recognize ts_>, we can first match the suffix, _>as an *ejective*, and then recognize ts as a consonant defined in $P_{base}$. The Algorithm 1 summarizes our approach, and the mapping of $f|_{P_{base}}$ is provided in Appendix.

## 2.2 SEQUENCE MODEL FOR ZERO-SHOT LEARNING

Zero-shot learning has rarely been applied to sequence prediction problems so far, in this section we describe a novel sequence model architecture for zero-shot learning. We adapt a modified ESZSL architecture from Romera-Paredes & Torr (2015). While the original architecture is devised to solve the classification problem with CNN(DECAF) features (Donahue et al., 2014), our model aims to optimize a CTC loss over a sequence model as shown in Figure 2. We note our architecture is a general model, and it can also be used for other sequence prediction problems in zero-shot learning.

Given the training set $\mathcal{S} = \{(\mathbf{x_n}, \mathbf{y_n}), n = 1...N\}$ where each input $\mathbf{x_n} \in \mathcal{X}$ is an utterance, and $\mathbf{y_n} \in \mathcal{Y}$ is the corresponding phonetic transcription. Suppose that $\mathbf{x_n} = (x_n^1, ..., x_n^T)$ is the input sequence where $x_n^t$ is the frame of time step $t$, and $T$ is the length of $\mathbf{x_n}$. Each frame $x_n^t$ is first projected into a feature vector $h_n^t \in \mathbb{R}^d$ in the acoustic space $\mathcal{H}$ with a Bidirectional LSTM model.

$$h_n^t = \theta(x_n^t; W_{\text{LSTM}}) \qquad (2)$$

where $W_{\text{LSTM}}$ is the parameter of Bidirectional LSTM model. We assume that our phoneme inventory consists of $z$ phonemes in the training set, each of them having a signature of $a$ attributes constructed as mentioned above. Then we can represent our attributes in a constant signature matrix $S \in [0,1]^{z \times a}$. Then, we transform $h_n^t$ into points in the phonetic space $\mathcal{P}$ with $V \in \mathbb{R}^{a \times d}$, and further into the phoneme logits $l_n^t$ with $S$.

$$l_n^t = SVh_n^t \qquad (3)$$

The logits $\mathbf{l_n} = (l_n^1, ..., l_n^T)$ is then combined with $\mathbf{y_n}$ to compute the CTC loss (Graves et al., 2006). Additionally, regularizing $V$ has been proved to be useful in the original ESZSL architecture (Romera-Paredes & Torr, 2015). So our target is to minimize following loss function.

$$\underset{V, W_{\text{LSTM}}}{\text{minimize}} \text{CTC}(\mathbf{x_n}, \mathbf{y_n}; V, W_{\text{LSTM}}) + \Omega(V) \qquad (4)$$

where $\Omega(V)$ is an simple L2 regularization. This objective can be easily optimized using standard gradient descent methods.

At inference stage, we usually consider a new language $L_{test}$ with a new phoneme inventory. Suppose that the new inventory is composed of $z'$ classes, then we can create a new signature matrix $S' \in [0,1]^{z' \times a}$, and estimate probability distribution of each phoneme $P_{acoustic}(p|x_n^t)$ from logits using $S'$ instead of $S$. Additionally, we expect the phoneme should distribute similarly to the phoneme distribution in $L_{test}$. We adjusted the posterior as follows.

$$P_{upm}(p|x_n^t) = P_{acoustic}(p|x_n^t)P(p|L_{test})^{\alpha} \tag{5}$$

where $\alpha$ is a hyperparameter, and $P(p|L_{test})$ is the prior distribution of $L_{test}$ which can be easily estimated from a small set of the text corpus in $L_{test}$ (we used 1000 sentences here). Finally we exploit greedy decoding to extract phoneme from the adjusted posterior $P_{upm}$(Graves et al., 2006).

## 3 EXPERIMENTS

### 3.1 DATASET

We prepared two datasets for this experiment. The training set consists of three English corpora, and the test set is composed of corpora from 20 languages. They are used by both our model and the baseline model described later. Details regarding each corpus and phonetic information about each language are provided at Table 1. Our model is trained with three English corpora because we tried to make the acoustic model robust to various channels and speech styles: TED (Rousseau et al., 2012) is the conference style, Switchboard (Godfrey et al., 1992) is the spontaneous conversation style and Librispeech is reading style (Panayotov et al., 2015). We note that 5 percent of the entire corpus was used as the validation set. The evaluation set were selected from a variety of languages: not only from well-resourced languages, but also low-resourced languages from Asia and Africa.

We list phoneme details for each language on the right sides in Table 1. All the phonemes were obtained using Epitran (Mortensen et al., 2018). The phonemes column shows the number of distinct phonemes in each language, the shared column is the number of shared phonemes with English, and the unseen column indicates the number of phonemes unseen in English, which means those unseen phonemes are not available in the training set.

### 3.2 EXPERIMENTAL SETTINGS

We used the EESEN framework for the acoustic modeling (Miao et al., 2015). Each corpus is first re-sampled to 8000Hz, and we extracted 40 dimension Mel-frequency cepstral coefficients (MFCCs) (Davis & Mermelstein, 1990) from each utterance, the length of each frame is 25ms, and the shift between two continuous frames is 10ms. All the transcripts were transcribed into phonemes with Epitran (Mortensen et al., 2018). We used a 5 layer Bidirectional LSTM model, each layer having 320 cells. The signature matrix is designed as we discussed above, and we used different signature matrices for different languages. We train the acoustic model with stochastic gradient descent, using a learning rate of 0.005. The prior distribution $P(p|L_{test})$ over phonemes is estimated by taking 1000 randomly selected sentences in the test language. Those 1000 sentences were removed from the test corpus. We fixed prior hyperparameter $\alpha = 1$ for all evaluations.

A standard English acoustic model is used as our baseline model (Miao et al., 2015). Phoneme distribution $P(p|x_n^t)$ is estimated with only Bidirectional LSTM $\theta(x_n^t; W_{\text{LSTM}})$. Then we decode phonemes with greedy decoding as we used in our approach. We use the same configuration of LSTM architecture as well as the training criterion. We also tried applying prior of target language to our baseline model as well. However, it was not helpful to improve the results because the phoneme inventory of English and target language are different, thus unique phonemes in target language cannot benefit from the prior distribution. As we focus on acoustic modeling in this work, we use phone error rate as the metric for evaluation.

### 3.3 RESULTS

Our results are summarized in Table 2. As it shows, our approach has consistently outperformed the baseline in terms of phone error rate. For example, the baseline achieves 77.3% phone error rate

| Language | Corpus Name | # Phonemes | # Shared | #Unseen |
|---|---|---|---|---|
| English | TED (Rousseau et al., 2012) | 38 | 38 | 0 |
| English | Switchboard (Godfrey et al., 1992) | 38 | 38 | 0 |
| English | Librispeech (Panayotov et al., 2015) | 38 | 38 | 0 |
| Amharic | ALFFA Amharic (Tachbelie et al., 2014) | 57 | 26 | 31 |
| Bengali | OpenSLR37 (Gutkin et al.) | 70 | 38 | 32 |
| Cebuano | IARPA-babel301b-v2.0b | 39 | 38 | 1 |
| Dutch | Voxforge | 49 | 34 | 15 |
| French | Voxforge | 42 | 33 | 9 |
| German | Voxforge | 37 | 25 | 12 |
| Haitian | IARPA-babel201b-v0.2b | 40 | 37 | 3 |
| Italian | Voxforge | 43 | 26 | 17 |
| Javanese | OpenSLR35 | 45 | 36 | 9 |
| Kazakh | IARPA-babel302b-v1.0a | 45 | 36 | 9 |
| Kurmanji | IARPA-babel205b-v1.0a | 27 | 23 | 4 |
| Lao | IARPA-babel203b-v3.1a | 39 | 21 | 18 |
| Mongolian | IARPA-babel401b-v2.0b | 50 | 18 | 32 |
| Russian | Voxforge | 48 | 22 | 26 |
| Sinhala | openSLR52 | 50 | 38 | 12 |
| Tagalog | IARPA-babel106b-v0.2g | 25 | 24 | 1 |
| Turkish | IARPA-babel105b-v0.4 | 40 | 38 | 2 |
| Swahili | ALFFA Swahili (Gelas et al., 2012) | 47 | 38 | 9 |
| Vietnamese | IARPA-babel107b-v0.7 | 47 | 37 | 10 |
| Zulu | IARPA-babel206b-v0.1e | 40 | 25 | 15 |

Table 1: Corpus of training set and test set used in the experiment. The acoustic model is trained with three English corpus, and tested on the 20 other languages below.

when evaluated with Amharic, and our approach obtained 68.7% in the same test set. For each language in our evaluation, we observed that we improved the phone error rate from 0.8% (French) to 21.7% (Italian) respectively. On average, the baseline model has 82.9% and our model get 9.9 % better phone error rate. The table also indicates one interesting relationship across different languages. The language which has close linguistic relationship with English tends to obtain better phone error rate. Dutch and German are classified in the West Germanic branch in the Indo-European language family like English. For Dutch, we obtained a 64.5% phone error rate and for German, a 61.6% phone error rate, which are the fourth and third best phone error rates among 20 languages. Additionally Italian and Russian belong to other branches of the Indo-European language family, but they also have very good phone error rate: 50.8% and 59.6% which are the first and second best phone error rates.

On the other hand, languages which are not a member of the Indo-European family tend to have worse phone error rate. The languages whose UPM phone error rate is worse than 80% are Lao, French, Zulu, Vietnamese. All languages except French are members of a different language family. Lao is in the KraDai language family, Zulu belongs to NigerCongo language family, and Vietnamese is a member of Austroasiatic language family. While French is a Indo-European language, its performance is much worse than other members of the language family. The reason is that French is a language known for its ambiguous orthographies (Ziegler et al., 1996), which makes it harder to make rule based grapheme-to-phoneme mappings. As our phonetic tool is designed as a rule-based grapheme-to-phoneme mappings, the prediction of French phonemes is difficult.

To further investigate the reason of improvements for our model, we computed the (phone) substitution error rate, shown in the two right columns of Table 2. It indicates that our model improves significantly over substitution errors: it goes down from 52.0% in baseline model to 38.2% in our model. The numbers shows that we have 13.8% improvement in substitution error rate. The improvements suggests that our model is good at recognizing confusing phonemes. Given the fact that largest improvement here is Italian and 3rd largest one is Russian, we can infer that the model is especially robust at recognizing phones in the same language family. We note that this number also

| Language | Baseline PER% | UPM PER% | Baseline Sub% | UPM Sub% |
|---|---|---|---|---|
| Amharic | 77.3 | 68.7 | 53.1 | 37.6 |
| Bengali | 82.6 | 67.3 | 62.9 | 38.2 |
| Cebuano | 88.7 | 72.1 | 48.9 | 27.2 |
| Dutch | 70.8 | 64.5 | 53.7 | 43.7 |
| French | 84.1 | 83.3 | 57.6 | 57.4 |
| German | 71.0 | 61.6 | 48.7 | 31.0 |
| Haitian | 84.5 | 74.4 | 54.5 | 41.5 |
| Italian | 72.5 | 50.8 | 56.4 | 27.5 |
| Javanese | 86.4 | 79.4 | 45.8 | 37.9 |
| Kazakh | 89.0 | 79.3 | 50.9 | 39.8 |
| Kurmanji | 91.3 | 79.1 | 46.6 | 30.5 |
| Lao | 87.9 | 85.6 | 55.2 | 54.8 |
| Mongolian | 87.6 | 79.8 | 43.7 | 28.6 |
| Russian | 73.4 | 59.6 | 55.2 | 33.9 |
| Sinhala | 76.3 | 71.2 | 55.3 | 45.5 |
| Tagalog | 86.2 | 70.9 | 46.6 | 25.7 |
| Turkish | 84.4 | 75.1 | 49.5 | 43.2 |
| Swahili | 83.8 | 70.8 | 45.3 | 27.0 |
| Vietnamese | 85.2 | 82.1 | 54.3 | 51.6 |
| Zulu | 95.8 | 85.1 | 56.5 | 41.9 |
| Average | 82.9 | **73.0** | 52.0 | **38.2** |

Table 2: Phone error rate (%PER) and phone substitution error rate (%Sub) of the baseline model, and our approach. Our model (UPM) outperforms the baseline model for all languages, by 9.9% (absolute) in phone error rate, and 13.8% in phone substitution error rate.

suggests addition and insertion error rate were getting worse by 3.9% than the baseline model, but our improvement in substitution error is enough to compensate for it.

| labels | phonemes |
|---|---|
| Gold Labels | b i k **s'** e **b ̱t** k a r i r a **h** \ a m **l ̱d** a **ts** \ a **l ̱d** a e ̱^ |
| baseline Prediction | b I k O r\ e j t h { m l E k E l i |
| UPM Prediction | b i k o e a **n ̱d l ̱d** a k a **l ̱d** e |

Table 3: A comparison of predictions in Bengali test set. The phonemes in bold font are unseen ones in English set, indicating our model's capability at predicting unseen phonemes.

To highlight the ability of our model to predict unseen phonemes, we show an example of Bengali in Table 3. According to Table 1, Bengali has the largest number of unseen phonemes among our test languages. Table 3 marks the unseen phonemes in English using bold font. The row of gold labels shows that target labels contain 7 unseen phonemes out of 22 phonemes. The baseline model was able to recognize part of seen phonemes such as the first phoneme /b/, however it fails to recognize all unseen phonemes due to its limitation. On the contrary, our model could correctly recognize two unseen phonemes /ḻd/ which aligned with two /ḻd/ phonemes in the gold labels. Random examples from other languages are available in Appendix.

## 4 RELATED WORK

We briefly outline several areas of related works, and describe their connections and differences with this paper. Firstly zero-shot learning has been applied to recognize unseen objects during training in the computer vision field. One line of works has focused on two-stage approach (Lampert et al., 2009; Palatucci et al., 2009; Lampert et al., 2014; Jayaraman & Grauman, 2014; Al-Halah

et al., 2016; Liu et al., 2011). Another line of approaches adapts multimodal models for zero-shot learning (Socher et al., 2013; Frome et al., 2013; Huang et al., 2012; Lei Ba et al., 2015; Rohrbach et al., 2011). Socher et al. (2013). However those works rarely mention speech recognition.

There has been growing interests in zero-resource speech processing (Glass, 2012; Jansen et al., 2013), most of the work focusing on tasks like acoustic unit discovery (Heck et al., 2017), unsupervised segmentation (Kamper et al., 2017), representation learning (Lake et al., 2014; Hermann & Goldwater, 2018), spoken term discovery (Dunbar et al., 2017; Jansen et al., 2013). These models are useful for various extrinsic speech processing tasks like topic identification (Kesiraju et al., 2017; Liu et al., 2017). However, those unsupervised concepts cannot be grounded to actual words or phonemes, hence making it impracticable to do speech recognition or acoustic modeling. The usual intrinsic evaluations that these zero resource tasks are tested on is ABX discriminability task (Dunbar et al., 2017; Heck et al., 2017) or the unsupervised word error rate (Ludusan et al., 2014; Kamper et al., 2017) which are good for quality estimates but not practical as they use an oracle or ground truth labels to assign cluster labels. Secondly these approaches demands a modest size of audio corpus of targeting language (e.g: 2.5h to 40h). In contrast, our approach assumes no audio corpus but a small set of text corpus to estimate phone prior. This is a reasonable assumption as text corpus is usually easier to obtain than audio corpus. The idea of decomposing speech into concepts was also discussed by Lake et al. (2014), where the authors propose a generative model to learn representations for spoken words which they then use to classify words with only one training sample available per word. Though this is also in the same line as the zero-resource speech processing papers, we feel the motivation behind the decomposition is very similar to this work.

The authors in Scharenborg et al. (2017) focus on a pipeline towards adaptation to a low resource language with little training data, they present an interesting method to map Dutch/English phonemes in the same space using an extrapolation approach to predict phones in English that are unseen in Dutch. Our work proposes a generic algorithm to recognize any unknown phones by decomposing them into its phone attributes. We have also shown that our approach is effective over 20 languages from different language families.

Another group of researchers explore adaptation techniques for multilingual speech recognition, especially for low resource languages. In these multilingual settings, the hidden layers are either HMM or DNN models which are shared by multiple languages, and the output layer is either language specific phone set or a universal IPA-based phone set (Tong et al., 2017; Vu & Schultz, 2013; Tüske et al., 2013; Thomas et al., 2010; Vu et al., 2014; Lin et al., 2009; Chen & Mak, 2015; Dalmia et al., 2018b). However predictable phonemes are restricted to the phonemes in the training set, thus they fail to predict unseen phonemes in the test set. In contrast, our model can predict unseen phonemes by taking advantage of their phonetic attributes.

Articulatory features have been shown to be useful in speech recognition under several situation. For example, articulatory features has been used to improve robustness under noisy and reverberant environment (Kirchhoff, 1998), compensate for crosslingual variability (Stüker et al., 2003b), improve word error rate in multilingual models (Stüker et al., 2003a), be beneficial for low resource languages (Müller et al., 2016), clustering phoneme-like units for unwritten languages (Müller et al., 2017). Those approaches generally treat articulatory features as additional feature for classifications, and did not provide a model to predict unseen phones.

## 5 Conclusion and Future Work

In this work, we propose the Universal Phonetic Model to apply zero-shot learning to acoustic models in speech recognition. Our experiment shows that it outperforms the baseline by 9.9 % phone error rate on average for 20 languages. While the performance of our approach is still not practical in the actual applications, it paves the way to tackle zero-shot learning of speech recognition with a new framework. We note that several approaches can be investigated on top of this framework. For instance, multilingual acoustic features can be used instead of standard ones (Hermann & Goldwater, 2018; Dalmia et al., 2018a). Label smoothing can be applied to regularize distributions of phonemes or attributes (Pereyra et al., 2017). The training objective can be replaced with discriminative training objectives such as MMI and sMBR (Veselỳ et al., 2013; Povey). Additionally, various encoder and decoder models can be explored with this framework (Chan et al., 2016; Zhou et al., 2018).

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

# A
## PHONETIC ATTRIBUTE FULL LIST

vowel consonant alveolar affricate alveolo-palatal approximant aspirated back bilabial breathy-voice central click close close-mid dental ejective epiglottal flap fricative front glottal implosive labial-palatal labial-velar labial labiodental lateral long nasal near-close near-open non-syllabic open open-mid palatal palatal-velar pharyngeal plosive postalveolar retroflex rounded schwa stop syllabic trill unrounded uvular velar voiced voiceless vowel coronal dorsal blank

# B
## PHONEMES AND ATTRIBUTES

| phoneme | attributes |
|---|---|
| a | vowel open front unrounded |
| b | consonant voiced bilabial stop labial |
| b_< | consonant voiced bilabial stop implosive labial |
| c | consonant voiceless palatal stop dorsal |
| d | consonant voiced alveolar plosive coronal |
| d' | consonant voiced retroflex plosive coronal |
| d_< | consonant voiced alveolar implosive coronal |
| e | vowel close-mid front unrounded |
| f | consonant voiceless labiodental fricative labial |
| pf | consonant voiceless labiodental fricative bilabial plosive labial |
| g | consonant voiced velar plosive dorsal |
| g_< | consonant voiced velar implosive dorsal |
| h | consonant voiceless glottal fricative |
| h\ | consonant voiced glottal fricative |
| i | vowel close front unrounded |
| j | consonant voiced palatal approximant dorsal |
| j\ | consonant voiced palatal fricative dorsal |
| k | consonant voiceless velar plosive dorsal |
| l | consonant voiced alveolar lateral approximant coronal |
| l' | consonant voiced retroflex lateral approximant coronal |
| l\ | consonant alveolar lateral flap coronal |
| m | consonant bilabial nasal labial |
| n | consonant voiced alveolar nasal coronal |
| n' | consonant retroflex nasal coronal |
| o | vowel close-mid back rounded |
| p | consonant voiceless bilabial plosive labial |
| p\ | consonant voiceless bilabial fricative labial |
| q | consonant voiceless uvular plosive |
| r | consonant alveolar trill coronal |
| r' | consonant retroflex flap coronal |
| r\ | consonant alveolar approximant coronal |
| r\' | consonant retroflex approximant coronal |
| s | consonant voiceless alveolar fricative coronal |
| s' | consonant voiceless retroflex fricative coronal |
| s\ | consonant voiceless alveolo-palatal fricative coronal |
| t | consonant voiceless alveolar plosive coronal |
| t' | consonant voiceless retroflex plosive coronal |
| t's' | consonant voiceless retroflex plosive fricative coronal |
| u | vowel close back rounded |
| v | consonant voiced labiodental fricative labial |
| w | consonant labial-velar approximant labial |
| x | consonant voiceless velar fricative dorsal |
| x\ | consonant voiceless palatal-velar fricative dorsal |
| y | vowel close front rounded |
| z | consonant voiced alveolar fricative coronal |
| z' | consonant voiced retroflex fricative coronal |
| z\ | consonant voiced alveolo-palatal fricative |

Table 4: phonemes and attributes

| phoneme | attributes |
|---------|------------|
| A | vowel open back unrounded |
| B | consonant voiced bilabial fricative labial |
| B\ | consonant bilabial trill labial |
| C | consonant voiceless palatal fricative |
| D | consonant voiced dental fricative coronal |
| E | vowel open-mid front unrounded |
| F | consonant labiodental nasal labial |
| G | consonant voiced velar fricative dorsal |
| G\ | consonant voiced uvular plosive dorsal |
| G_< | consonant voiced uvular implosive |
| H | consonant labial-palatal approximant labial |
| H\ | consonant voiceless epiglottal fricative |
| I | vowel near-close front unrounded |
| I\ | vowel near-close central unrounded |
| J | consonant palatal nasal |
| J\ | consonant voiced palatal plosive |
| J_< | consonant voiced palatal implosive |
| K | consonant voiceless alveolar lateral fricative coronal |
| K\ | consonant voiced alveolar lateral fricative coronal |
| L | consonant palatal lateral approximant |
| L\ | consonant velar lateral approximant dorsal |
| M | vowel close back unrounded |
| M\ | consonant velar approximant dorsal |
| N | consonant velar nasal dorsal |
| N\ | consonant uvular nasal dorsal |
| O | vowel open-mid back rounded |
| O\ | consonant bilabial click labial |
| P | consonant labiodental approximant labial |
| Q | vowel open back rounded |
| R | consonant voiced uvular fricative |
| R\ | consonant uvular trill |
| S | consonant voiceless postalveolar fricative coronal |
| T | consonant voiceless dental fricative coronal |
| U | vowel near-close back rounded |
| U\ | vowel near-close central rounded |
| V | vowel open-mid back unrounded |
| W | consonant voiceless labial-velar fricative labial |
| X | consonant voiceless uvular fricative |
| X\ | consonant voiceless pharyngeal fricative |
| Y | vowel near-close front rounded |
| Z | consonant voiced postalveolar fricative coronal |

Table 5: phonemes and attributes

| phoneme | attributes |
|---|---|
| : | long |
| ' | palatal |
| @ | vowel close-mid open-mid central rounded unrounded |
| { | vowel near-open front unrounded |
| } | vowel close central rounded |
| 1 | vowel close central unrounded |
| 2 | vowel close-mid front rounded |
| 3 | vowel open-mid central unrounded |
| 4 | consonant alveolar flap coronal |
| 5 | consonant velar alveolar lateral approximant coronal dorsal |
| 6 | vowel near-open central |
| 7 | vowel close-mid back unrounded |
| 8 | vowel close-mid central rounded |
| 9 | vowel open-mid front rounded |
| & | vowel open front rounded |
| ? | consonant glottal stop |
| ?\ | consonant voiced pharyngeal fricative |
| —\ | consonant dental click coronal |
| —\—\ | consonant alveolar lateral click coronal |
| =\ | consonant palatal click |
| _< | implosive |
| _> | ejective |
| ~ | nasal |
| _h | aspirated |
| _w | labial |
| _t | breathy-voice |
| _^ | non-syllabic |
| _d | dental coronal |
| ts | consonant voiceless alveolar affricate coronal |
| dz | consonant voiced alveolar affricate coronal |
| dz\ | consonant voiced alveolo-palatal affricate coronal |
| dZ | consonant voiced postalveolar affricate coronal |
| dZ\ | consonant voiced alveolo-palatal affricate coronal |
| tS | consonant voiceless postalveolar affricate coronal |
| ts\ | consonant voiceless alveolo-palatal affricate coronal |
| s\ | consonant voiceless palatal fricative dorsal |
| tK | consonant voiceless alveolar lateral affricate coronal |

Table 6: phonemes and attributes

| labels | phonemes |
|---|---|
| Amharic Gold | l n l s l r a w t @ S @ n @ **k_>** o r @ |
| Amharic Baseline | V n t V r\ V V t e S V n o p k w e t |
| Amharic UPM | n t a t e j a n o **k_>_w** w a t |
| Bengali Gold | e b i **s'** **e_^** o t i k e |
| Bengali Baseline | I b I dZ o j d i g e j |
| Bengali UPM | e b i **dz\** o i g e |
| Cebuano Gold | a w l a g i a N |
| Cebuano Baseline | V i j { a j m n A t b i h |
| Cebuano UPM | o n a i j a j m n a t b i h a |
| Dutch Gold | t P **e: e:** p l Y s t r **i: e:** I s v E j f |
| Dutch Baseline | dZ e j p I z d V t i I S a j d |
| Dutch UPM | dZ H e H p H I z d V t **i:** I S **a:** j d |
| French Gold | i l E s t s y s p E n d y d @ d @ l a l E g j **O~** |
| French Baseline | i l E t j i s p A w n d i D V r\ A b A D V l E l I Z V n V n r\ |
| French UPM | H i l E t H i s H a H n d l H i d H **A~** H H **A~** n E l E l i d **A~** H n **A~** n E |
| German Gold | i m j **a:** r v u r d @ d @ r a l g **o:** r i t m u s b e S r **i:** e b @ n |
| German Baseline | m i j A r\ n O t V n V n D V s i w V n s i p s I k w o |
| | d D E r\ A g I d m o w s t b I S r\ i v V n |
| German UPM | m i a n a t i n i n d t s i m i n s i s i k w o t d **e:** a g t m o s t b i s l i v i n |
| Haitian Gold | w i m p a k **O~** n s a p u m w f e s i m t e b **O~** n v i w **O~** m s a a s e m m Z |
| Haitian Baseline | w i b A k V n b A t I n s V s V m d E b V n A l dZ o r\ m A |
| | T O f j u s V m E m V z O l |
| Haitian UPM | w i p a k e m p a t m s e s e m d e p e n a l j o l m **O~** s a j **O~** s e m e m i **O~** l |
| Italian Gold | g u a r d a b e n e t u k e h a i l a v i s t a p i u a k u t a d i m e |
| Italian Baseline | w V V b i n i t u k e j l { v i s t V p j O r\ k u t V d i m e j |
| Italian UPM | v e t a b e n i t u k e l e v i s t a p i o k u t a d i m e n |
| Javanese Gold | i j a i k i i j a i j a **d_d** e w i |
| Javanese Baseline | V E j { j { I r\ e |
| Javanese UPM | i a J **d'** a a i |
| Kazakh Gold | A l l w U A l **i_^** @ j k w m A s s A l A w m |
| Kazakh Baseline | o w w E l |
| Kazakh UPM | A w w **i_^** l H A |

Table 7: A list of random samples from the first 10 test languages. Bold phonemes are unseen phonemes in English

