# OpenReview forum: "Zero-shot Learning for Speech Recognition with Universal Phonetic Model"
_ICLR.cc/2019/Conference_

### Official Review · AnonReviewer2 · 2018-11-01
**Claims of being first not completely justified**

**Rating:** 5
**Confidence:** 4

**Review:**

Overview:

This paper proposed an approach for zero-shot phoneme recognition, where it is possible to recognise phonemes in a target language which has never been seen before. Rather than just training a phoneme recogniser directly on background data and then applying it to unseen data, phonetic features are first predicted, allowing phonemes not in the source language set to be predicted.

Main strengths:

The paper's main strength lies in that this is a very unexplored area that could assist in the development of speech technology where it is currently not possible. The proposed model (Section 2) has also not been considered in prior work.

Main weaknesses:

The paper's main weakness is in some of its claims and that it misses some very relevant literature. Detailed comments together with a minimal list of references are given below (but I would encourage the authors to also read a bit more broadly). But in short I do not think it is that easy to claim that this is the first paper to do zero-shot learning on speech; many of the zero-resource studies where unlabelled audio is used could be seen as doing some for of zero-shot matching. Specifically [5] is able to predict unseen phoneme targets.  Multilingual bottleneck features can be applied to languages that have never been seen before [2], and the output of phoneme recognisers trained on one language have long been applied to get output on another unseen language. The first one-shot learning speech paper [4] (to my knowledge) is also not mentioned at all. The approach in the paper also still relies on some text data from the target language; if this then can be described as "zero-shot" learning, then I think many of these previous studies c also make this claim.

Overall feedback:

There is definitely value in this work, but it should be much better situated within the broader literature. Below I give some editorial suggestions and also outline some suggestions for further experiments.

Detailed comments, suggestions and questions:

- Abstract: It would be useful to have some details of the "baseline model" here already, especially since it is such a new task.
- Introduction: "... but they can hardly predict phones or words directly due to their unsupervised nature." This is a strong statement that maybe requires more justification. On the one hand, the statement is true, and the high word error rates in e.g. [3] can be cited. On the other hand, it has been shown that at the phone-distinction level, these models perform quite well and sometimes outperform supervised models [1]. Since this paper also considers phone error rate as a metric, I think care should be taken with such statements.
- Introduction: "While zero-shot learning has attracted a lot of attention in *the* computer vision community, this setup has hardly been studied in speech recognition research especially in acoustic modeling." Definitely look at some of the studies mentioned below, and also [4] specifically.
- "However, we note that our model can be combined with a well-resourced language model to recognize words." How would this be done, since I think this is actually quite a challenging task.
- Section 2: "... useful the original ESZSL architecture ..." -> "... useful in the original ESZSL architecture ..."
- Section 2.2: I assume the small text corpus is at the phone level (and not characters directly)? This should be clarified, and it could raise the question of whether this approach is truly "zero-shot".
- Section 3.2: "We used EESEN framework ..." -> "We used the EESEN framework ..."
- Section 4: You could look at the recent work in [2], which uses multilingual bottleneck features trained on 10 languages and applied to multiple unseen languages. It would be interesting to also train your approach on multiple languages instead of only English.

Missing references:

1. M. Heck, S. Sakti, and S. Nakamura, "Feature Optimized DPGMM Clustering for Unsupervised Subword Modeling: A Contribution to Zerospeech 2017," in Proc. ASRU, 2017.
2. E. Hermann and S. J. Goldwater, "Multilingual bottleneck features for subword modeling in zero-resource languages," in Proc. Interspeech, 2018.
3. H. Kamper, K. Livescu, and S. Goldwater, An embedded segmental k-means model for unsupervised segmentation and clustering of speech," in Proc. ASRU, 2017.
4. B. M. Lake, C.-Y. Lee, J. R. Glass, and J. B. Tenenbaum, "One-shot learning of generative speech concepts," in Proc. CogSci, 2014.
5. O. Scharenborg, F. Ciannella, S. Palaskar, A. Black, F. Metze, L. Ondel, and M. Hasegawa-Johnson, "Building an ASR system for a low-resource language through the adaptation of a high-resource language asr system: Preliminary results,"in Proc. ICNLSSP, 2017.

Edit: Based on the rebuttal I've changed my rating from 4 to 5.

---

> ### Author Response · Authors · 2018-11-21
> **Reply to reviewer 2**
>
> Thank you for detailed comments and references ! We will use them to enhance our paper by providing more discussion of related works. We discuss several points that distinguish our paper from the suggested references here -
>
> 1.2.3: Those papers are works in zero resource speech recognition. As you suggested, we will discuss more about the connection between our work and those papers. Those zero resource works assume that no transcribed labels are available but a lot of audio data is provided for the target language. In contrast, our work assumes that both transcribed labels and audio are not available for the test language, but we use a limited amount of text sentences instead.
>
> 4. We were unaware of this work and will update our related work. The work is applying one-shot learning to speech recognition by proposing a generative model. As its name suggests, the work was trying to classify words with only one training sample available per word. Our work is different from this one because we are using no training speech data for the target corpus.
>
> 5. Thanks for pointing us to this work. We investigated further on this paper and also talked to some of the authors of the paper. We agree that the motivation behind this work is similar but it is limited to some extent. This work proposes an extrapolation approach to predict phones for low resource languages, however, the extrapolation mapping is done manually. Additionally the evaluation is carried out on Dutch/English pair which is similar in terms of their phonetics and language family. It does not show whether the approach will work for language pairs from unknown/unrelated linguistic groups . In contrast, our work proposes a generic algorithm to recognize any unknown phones by decomposing them into its phone attributes. We have shown that our approach is effective over 20 languages from different language families.

---

> > ### Comment · AnonReviewer2 · 2018-11-28
> > **Update**
> >
> > Based on the rebuttal I've changed my rating from 4 to 5.

---

> > > ### Author Response · Authors · 2018-11-29
> > > **Reply**
> > >
> > > Thanks for updating your rate. After your rating update on 26th Nov we edited the paper and also added in some more discussions regarding the points from your comments that we may have missed before.

---

> ### Author Response · Authors · 2018-11-27
> **Paper revision comment for reviewer 2**
>
> Thanks again for the detailed comments and references! We have tried to answer almost all the detailed comments, suggestions and questions that you mentioned in our revised paper.
>
> In detail the changes that we made and some comments -
> 1. Abstract now mentions what the baseline model is.
> 2. Corrected the strong statements mentioned in the introduction with proper justification and added in the missing references. We added a discussion about zero resource speech processing tasks in related work and introduction referencing 1,2,3 and some more work.
> 3. Thanks for pointing out to reference 4 and 5! We have talked about them in detail in related work section.
> 4. Language model can be integrated using WFST decoding by assuming that our language model is well resourced and considering the words seen during language model training as the words being predicted during decoding.
> 5. Thanks a lot for pointing out the grammatical errors we have now fixed them.
> 6. Yes we agree multilingual bottleneck features would be helpful, but the problem of having “unseen” phones would still exist no matter how many languages we add. This paper focuses on solving this issue where the model has seen only English phones during training. Infact this idea would lead to a better model as it would have better coverage of the phonemes and it would be a good extension towards this paper, hence mentioned as part of future work.

---

### Official Review · AnonReviewer3 · 2018-11-01
**Interesting but not good enough!**

**Rating:** 4
**Confidence:** 4

**Review:**

This paper proposes to train a Universal Phonetic Model for building speech recognition for new languages without any training data. It suggests to use X-SAMPA to map phones from all the languages into a single phonetic space. The prediction models are designed to first predict the phonetic features and then the phones depending on the target language.
Overall , the paper is quite clear written.
- Strengthens:
+ It observed overall improvements for all the target languages.

- Weaknesses:
+ The idea and the proposed model are not novel.
+ All the baseline systems have relative high phone error rates.
+ The authors claimed to have a universal phonetic model but actually the model was trained only with English data. Therefore, experimental setup could be improved. In my opinion, it makes more sense to define a bunch of resource-rich languages as source and then train a real universal phonetic model.
+ Overall, this paper lacks an analysis what are exactly improved and why the improvements for some target languages are larger than for the others.

---

> ### Author Response · Authors · 2018-11-21
> **Reply to reviewer 3**
>
> Thank you for your valuable comments !
>
> We think that the term “Universal Phonetic Model” might have confused the reviewer. We are sorry about that. The problem that we want to address is the task of zero-shot learning for speech recognition, which consist of learning an acoustic model without any resources for a given target language. We call our model “Universal Phonetic Model”, because it has the ability to predict any phoneme, even the ones that are not present during training (therefore it covers a “universal” set). We achieve this by decomposing the phone label into its phone attributes.
>
> One of the weakness that has been pointed out is that the idea and model is not novel. However, we did not find any works that attempt the same problem with a similar model. It is possible we are unaware of related work, it would be helpful if the reviewer can give some references so that we can investigate further. Most of the work on zero-shot in speech community that we found only identified “similar” speech concepts or sounds, but could not ground them to phone labels making it hard to do speech recognition. Similarly, the idea of decomposing sounds into articulatory features is old, but our work presents the first approach that actually decomposes sounds into “universal” articulatory features and recognizes speech in unseen languages using such representations.
>
> As we mentioned to AnonReviewer1, we agree that our baseline model has too high a phone error rate to be usable in practice. Unfortunately this is what the current CTC acoustic models provide for the task of zero-shot speech recognition. Both the baseline and UPM models had practical and competitive phoneme error rate in the test data of the 3 english datasets that were used during training. However, we do believe that some of the performance reduction when using this model cross-lingually and cross-domain could because our input features are not robust against acoustic domain mismatch. We are currently re-running the experiments with a new set of input features proposed in [1], and first results indicate that we can get even better improvements in the same settings, and on top of a much improved baseline. We believe this is due to the stronger (noise robust and domain invariant) overall baseline allowing for a better sharing of the linguistically informative information across languages, and we are working towards applying this idea to all the experiments including baseline, so that an updated version of the paper will again be consistent.
>
>
> We do agree that a better universal phoneme recognizer can be built by training on even more languages. But we believe that our experiments show that problem we want to address here, specifically the ability to predict unseen phonemes, in a zero-shot speech recognition scenario, can be tackled with the proposed method. Using more training languages would reduce appearance of unknown phonemes, but there will still almost always be at least a few unseen phones, which we show our model is effective in reducing..
>
> [1] S.Dalmia, X. Li, F. Metze and AW Black, “Domain Robust Feature Extraction for Rapid Low Resource ASR Development”, in Proc SLT 2018, https://arxiv.org/pdf/1807.10984.pdf

---

### Official Review · AnonReviewer1 · 2018-11-02

**Rating:** 7
**Confidence:** 4

**Review:**

This paper presents an approach to address the task on zero-shot learning for speech recognition, which consist of learning an acoustic model without any resources for a given language. The universal phonetic model is proposed, which learns phone attributes (instead of phone label), which allows to do prediction on any phone set, i.e. on any language. The model is evaluated on 20 languages and is shown to improve over a baseline trained only on English.

The proposed UPM approach is novel and significant: being able to learn a more abstract representation for phones which is language-independent is a very promising lead to handle the problem of ASR on languages with low or no resources available.

However, the results are the weak point of the paper. While the results demonstrate the viability of the approach, the gain between the baseline performance and the UPM model is quite small, and it's still far from being usable in practice.

To improve the paper, the authors should discuss the future work, i.e. what are the next steps to improve the model.

Overall, the paper is significant and can pave the way for a new category of approaches to tackle zero-shot learning for speech recognition. Even if the results are not great, as a first step they are completely acceptable, so I recommend to accept the paper.

Revision:
The approach of using robust features is interesting and promising, as well as the idea of training on multiple languages. Overall, the authors response addressed most of the issues, therefore I am not changing my rating.

---

> ### Author Response · Authors · 2018-11-21
> **Reply to reviewer 1**
>
> We appreciate your time reviewing our paper ! Thank you for your encouraging comments and remarks. We agree that our baseline model has too high a phone error rate to be usable in practice. We believe that this is because the input features currently being used are not robust against acoustic domain mismatch. We are currently re-running the experiments with a new set of input features proposed in [1], and first results indicate that we can get even better improvements in the same settings, and on top of a much improved baseline. We believe this is due to the stronger (noise robust and domain invariant) overall baseline allowing for a better sharing of the linguistically informative information across languages, and we are working towards applying this idea to all the experiments including baseline, so that an updated version of the paper will again be consistent.
>
> [1] S.Dalmia, X. Li, F. Metze and AW Black, “Domain Robust Feature Extraction for Rapid Low Resource ASR Development”, in Proc SLT 2018, https://arxiv.org/pdf/1807.10984.pdf

---

> > ### Comment · AnonReviewer2 · 2018-11-22
> > **Clarification**
> >
> > Will these additional experiments be included in the revised manuscript?

---

> > > ### Author Response · Authors · 2018-11-27
> > > **Reply**
> > >
> > > Thanks for your question. We are running some new experiments with the domain robust acoustic feature. Our initial experiment on a reduced dataset suggests that these features have potential to improve performance by about 5 percent, but due to our computational limitations we could not complete our experiments on the whole dataset and have mentioned it as part of the future work.

---

> ### Author Response · Authors · 2018-11-27
> **Paper revision comment for reviewer 1**
>
> Thanks again for your comments, as suggested we revise the paper and mention a few potential works that can be extended on top of the proposed framework. For instance, label smoothing might be a useful technique to regularize attribute distribution or phone distribution [1], we can also increase the coverage of our phonemes by training the model on more diversity of languages or by training them with better features.
>
> [1] Pereyra, Gabriel, et al. "Regularizing neural networks by penalizing confident output distributions." arXiv preprint arXiv:1701.06548 (2017).

---

### Meta-Review · Area_Chair1 · 2018-12-14
**Good first step, but error rates are too high**

**Confidence:** 5
**Recommendation:** Reject

**Metareview:**

This paper studies the really hard problem of zero-shot learning in acoustic modeling for languages with limited resources, using data from English. Using a novel universal phonetic model, the authors show improvements compared to using an English model for 20 other languages in phone recognition quality.

Strengths
- Reviewers agree that the problem is an important one, and the presented ideas are novel.
- Universal phonetic model to represent phones in any language is interesting.

Weaknesses
- The results are really weak, to the point that it is unclear how effective or general the techniques are. The work is an interesting first step, but is not developed enough to be accepted at this point.
- The universal phonetic model being trained only in English might affect generalizability to languages that do not share phonetic characteristics. The authors agree partly, and argue that the method already addresses some issues since the model can already represent unseen phones. But, coupled with the high phone error rates, it is still unclear how appropriate the technique will be in addressing this issue.
- Novelty: Although the idea of mapping phones to attributes, and using those for ASR is not novel (e.g., using articulatory features), application for zero-shot learning is. The work assumes availability of a small text corpus to learn phone-sequence distribution, so is similar to other zero-resource approaches that assume some data (audio, as opposed to text) is available in the new language.

This paper presents interesting first steps, but lacks sufficient experimental validation at this point. Therefore, AE recommendation is to reject the paper. I encourage the authors to improve and resubmit in the future.